# Genetic Characteristics of Wuxiang Virus in Shanxi Province, China

**DOI:** 10.3390/v16010103

**Published:** 2024-01-10

**Authors:** Yuke Zheng, Xiaodong Tian, Ruichen Wang, Xiaohui Yao, Weijia Zhang, Qikai Yin, Fan Li, Kai Nie, Qianqian Cui, Songtao Xu, Shihong Fu, Hao Li, Jingxia Cheng, Huanyu Wang

**Affiliations:** 1Chinese Center for Disease Control and Prevention, Beijing 102206, China; zhengyk1231@163.com; 2Shanxi Province Center for Disease Control and Prevention, Taiyuan 030012, China; 15935113506@163.com; 3National Key Laboratory of Intelligent Tracking and Forecasting for Infectious Diseases, National Institute for Viral Disease Control and Prevention, Chinese Center for Disease Control and Prevention, Beijing 102206, China; wangrc96@163.com (R.W.); xiaohuiyao1995@163.com (X.Y.); zwj_0308@126.com (W.Z.); yinqk@ivdc.chinacdc.cn (Q.Y.); lifan@ivdc.chinacdc.cn (F.L.); niekai@ivdc.chinacdc.cn (K.N.); cuiqq@ivdc.chinacdc.cn (Q.C.); xust@ivdc.chinacdc.cn (S.X.); fush@ivdc.chinacdc.cn (S.F.); 4Jiangsu Co-Innovation Center for Prevention and Control of Important Animal Infectious Diseases and Zoonoses, College of Veterinary Medicine, Yangzhou University, Yangzhou 225009, China

**Keywords:** Wuxiang virus, sandfly-borne *Phlebovirus*, sandfly, genetic characteristics

## Abstract

Wuxiang virus (WUXV) is the first sandfly-borne *Phlebovirus* isolated from *Phlebotomus chinensis* collected in China and has been established as a consistent viral presence in the local sandfly populations of both Wuxiang County and Yangquan City. However, its distribution in the Shanxi Province remains unclear. In this study, three novel WUXV strains were isolated from sandflies collected from Jiexiu City, Shanxi Province, China, in 2022. Subsequently, whole-genome sequences of these novel strains were generated using next-generation sequencing. The open reading frame (ORF) sequences of the WUXV strains from the three locations were subjected to gene analysis. Phylogenetic analysis revealed that WUXV belongs to two distinct clades with geographical differences. Strains from Wuxiang County and Yangquan City belonged to clade 1, whereas strains from Jiexiu City belonged to clade 2. Reassortment and recombination analyses indicated no gene reassortment or recombination between the two clades. However, four reassortments or recombination events could be detected in clade 1 strains. By aligning the amino acid sequences, eighty-seven mutation sites were identified between the two clades, with seventeen, sixty, nine, and one site(s) in the proteins RdRp, M, NSs, and N, respectively. Additionally, selection pressure analysis identified 17 positively selected sites across the entire genome of WUXV, with two, thirteen, one, and one site(s) in the proteins RdRp, M, NSs, and N, respectively. Notably, sites M-312 and M-340 in the M segment not only represented mutation sites but also showed positive selective pressure effects. These findings highlight the need for continuous nationwide surveillance of WUXV.

## 1. Introduction

Wuxiang virus (WUXV) is a sandfly-borne virus belonging to the *Phlebovirus* genus, *Phenuiviridae* family, *Bunyavirales* order. According to the latest International Committee on Taxonomy of Viruses classification guidelines, the *Phlebovirus* genus includes 67 different viral species that can be transmitted by sandflies, mosquitoes, ticks, or other arthropods [1,2]. Sandflies, the primary vectors of phleboviruses, have a blood-sucking habit and feed on a variety of sources, including humans, dogs, chickens, and sheep. Their attack is silent, and they are crepuscular–nocturnal but some may bite during daylight. Sandflies are distributed throughout the world in tropical and subtropical, arid and semi-arid, and temperate zones [3], while sandfly-borne phleboviruses and the epidemic areas of virus-related diseases are mainly distributed in Mediterranean countries. However, with the escalation of travel, transportation, and international trade, non-Mediterranean countries face an ongoing and elevated risk of importing cases of phlebovirus infections. This is exemplified by the report of sandfly-borne phlebovirus infections in regions such as the Middle East, Central Asia, and East Africa [4,5,6]. Most phleboviruses can cause a febrile syndrome, commonly known as sandfly fever, and a few phleboviruses, such as the Toscana virus, are associated with neurological diseases in humans, particularly during the summer months [7,8]. Phleboviruses are a significant zoonotic and pandemic threat to both human and animal populations due to the global presence of both the hosts and the arthropod vectors. Therefore, developing policies to prevent and control the diseases they cause is necessary [9].

WUXV, the first *Phlebovirus* isolated from sandflies in China, was isolated from *Phlebotomus chinensis* collected from Wuxiang County, Shanxi Province, China, in 2018 [10]. Only mosquito- or tick-borne phleboviruses, such as the Rift Valley Fever Virus (RVFV) and Guertu virus (GTV), were reported in China until the discovery of WUXV [11,12]. In the same year as its initial discovery in Wuxiang County, WUXV was also isolated from Yangquan City in Shanxi Province [13]. Subsequent investigations revealed the virus’s stable existence in the indigenous sandfly populations of both locations [14,15,16]. The detection of neutralizing antibodies against WUXV and seroepidemiological studies suggest there are widespread infections of WUXV in mammals (dogs, chickens) and humans from the regions where these viruses have been isolated, indicating that WUXV can infect both humans and animals [16,17]. In addition, WUXV exhibits the capability to replicate within the brain tissue of newborn suckling mice, resulting in pronounced neurological manifestations such as failing to consume milk, convulsion and rigor of the limbs, lateral lying position, and negative turning reflex. This viral infection has been shown to have an impact on the morbidity and mortality rates among suckling mice [13]. Furthermore, the development of two molecular detection methods for WUXV has revolutionized the screening process by enabling the rapid identification of WUXV nucleic acids in a large number of samples, thus facilitating rapid detection of the virus [18,19]. Additionally, the susceptibility of BHK-21, MDCK, and PK-15 cell lines to WUXV provides vital information for future ex vivo studies on this virus [20].

WUXV is characterized by spherical virions, ranging in size from 80 to 120 nm, encapsulated by a lipid bilayer. Its genome consists of three single-stranded negative-sense RNA segments: large (L), medium (M), and small (S) [10]. The L segment encodes an RNA-dependent RNA polymerase (RdRp), which is responsible for the replication and transcription of viral RNA in the cytoplasm of infected cells [21]. The M segment encodes a glycoprotein precursor that is cleaved into the nonstructural protein NSm, the glycoprotein N (Gn), and the glycoprotein C (Gc) by cellular proteases during translation. Gn and Gc are the targets of specific neutralizing antibodies and are responsible for attachment and membrane fusion, respectively, which are required for host cell entry [22,23,24]. The S segment utilizes an ambisense strategy, encoding two proteins (N and NSs). For *Phlebovirus*, NSs is recognized as the primary virulence factor and plays a crucial role in the virus’s ability to evade the host’s innate immunity [25,26]. However, there are almost no studies on the genetic characteristics of the WUXV genome.

In this study, three WUXV strains were isolated from *P. chinensis* collected from Jiexiu City, Shanxi Province, in 2022. The open reading frame (ORF) sequences of the novel and previously isolated WUXV strains were subjected to genetic analysis to assess the phylogeny and evolution of WUXV and to determine its genetic characteristics.

## 2. Materials and Methods

### 2.1. Sample Collection

Two villages in Jiexiu City, Shanxi Province, China were selected for sandfly sampling on 29 July 2022: Baishui village (111°44′56″ E, 37°1′23″ N) and Guobi village (111°47′27″ E, 37°0′19″ N) (Figure 1). The sandflies were collected using insect collectors (MM200; Guangzhou Changsheng Chemical Co., Ltd., Guangzhou, China). The insect collectors were placed in sheep pens at each sampling point from 6 p.m. to 7 a.m. The collected sandflies were placed in a low-temperature refrigerator for at least 20 min, classified according to insect morphology and collection environment, and stored in liquid nitrogen until further testing [10,27].

### 2.2. Cells and Viruses

Golden hamster kidney cells (BHK-21 cells) were cultured in 89% Minimum Essential Medium (MEM, Gibco, Thermo Fisher Biochemical Products [Beijing] Co., Ltd., Beijing, China) supplemented with 10% fetal bovine serum (FBS; Sigma-Aldrich, Merck KGaA, Darmstadt, Germany), 1% penicillin, and streptomycin (100 U/mL). *Aedes albopictus* cells (C6/36 cells) were cultured in 89% Roswell Park Memorial Institute (RPMI) 1640 medium (Gibco), supplemented with 10% FBS (Sigma), 1% penicillin, and streptomycin (100 U/mL). BHK-21 cells were cultured at 37 °C and 5% CO_2_, while C6/36 cells were cultured at 28 °C. Twenty-eight previously isolated WUXV strains (eleven from Wuxiang County and seventeen from Yangquan City) were maintained in our laboratory for subsequent sequencing (Appendix A).

### 2.3. Virus Isolation and Identification

The sandflies were divided into 45 groups of approximately 30 sandflies each and placed into a grinding tube with 1 mL of grinding solution (95% MEM [Gibco], 5% penicillin, and streptomycin [100 U/mL]). Samples were ground using TissueLyser Ⅱ (Qiagen, Hilden Germany) and centrifuged at 4 °C and 12,000 rpm for 30 min. Subsequently, 75 uL of the supernatant was inoculated into 24-well plates (Corning Inc., New York, NY, USA) of BHK-21 or C6/36 cells seeded at 80% density. After inoculation, BHK-21 and C6/36 cells were grown continuously at 37 °C and 28 °C, respectively. The cytopathic effect (CPE) was monitored every 12 h under a microscope, and viral RNA was extracted from the infected BHK-21 or C6/36 cell cultures using the Ex-DNA/RNA virus 4.0 Kit (T324; Xi’an Tianlong Science&Technology Co., Ltd., Xi’an, China). The HiScript III U+ One Step qRT-PCR Probe Kit (Nanjing Vazyme Biotech Co., Ltd., Nanjing, China) was used to detect the nucleic acids of WUXV. The sequences of specific primers and probes for WUXV used in this study have been described previously [16]. Samples not displaying CPEs underwent three generations of blind passage through BHK-21 and C6/36 cells. Those failing to induce CPE were excluded [10,27]. 

### 2.4. Viral Genome Sequencing

RNA-Seq libraries were constructed using the VAHTS Universal V8 RNA-Seq Library Prep Kit (Vazyme). The libraries were sequenced by Beijing Dia-up Biotech Ltd. (Beijing, China) on a Nova-Seq platform in paired-end 150 bp mode. The generated raw reads underwent further processing, involving the removal of reads containing adapters, poly-N, and those with low sequencing quality. The clean paired-end reads were then mapped to the reference sequences (WUXV strain SXWX1813-2 (GenBank accession: L segment, MN454526; M segment, MN454527; S segment, MN454528)) using CLC Genomics Workbench v20.0.4 to generate a consensus sequence for each segment.

### 2.5. Gene Analysis

The WUXV dataset used for gene analysis consisted of ORF sequences of WUXV strains isolated from three locations in Shanxi Province (Wuxiang County, Yangquan City, and Jiexiu City; Appendix A). *Phlebovirus corfouense* strain Pa Ar 814 (GenBank accession numbers: L segment, KR106177; M segment, KR106178; S segment, KR106179) was used as an outgroup for phylogenetic analysis (Appendix A). MegAlign v7.1.0 was employed for sequence alignment and homology analysis. MEGA v7.0.26 was used for the phylogenetic analysis based on the neighbor-joining method, with a bootstrap value of 1000. The generated phylogenetic trees were identified using Chiplot (https://www.chiplot.online/normalTree.html, accessed on 6 November 2023) [28]. RDP4 v4.100 was used to screen reassortment or recombination events based on seven detection algorithms, and only reassortment or recombination events supported by at least five algorithms with a *p*-value < 0.01 were accepted [29]. Accepted reassortment or recombination events were further verified using SimPlot v3.5.1. HyPhy v2.5.8 was used for selection pressure analysis of the WUXV genome, with the FEL, FUBAR, MEME, and SLAC methods applied with default parameters [30,31,32]. The tertiary structure of the M protein in each clade was predicted using AlphaFold2 (https://cryonet.ai/af2/, accessed on 13 November 2023) and further aligned using PyMOL v2.5.5.

### 2.6. Morphological and Genetic Identification of Sandfly Species

The collected sandflies were dried and subsequently completely infiltrated with 75% ethanol. Prior to further processing, their body surfaces were carefully depilated by means of shaking. The sandflies were then rinsed with double-distilled water before the head and abdominal Sections 8–10 were cut off under a microscope and immersed in 10% KOH solution at 37 °C for 20 h. After being thoroughly washed with double distilled water, the cibarium and pharynx were dissected using a fine needle and subsequently observed and photographed under a microscope. The sandfly species were identified according to their morphological characteristics [33]. The DNA of the sandflies was extracted and subjected to PCR to amplify the cytochrome C oxidase I (COI) gene, using the forward primer LCO1490 (5′-GGTCAACAAATCATAAAGATATTGG-3′) and the reverse primer HC02198 (5′-TAAACTTCAGGGTGACCAAAAAATCA-3′) [13]. The nucleotide sequences of the PCR products were determined by Sanger sequencing. Then, a BLAST search against the GenBank database was performed to identify the species of the sandflies.

## 3. Results

### 3.1. Sandfly Collection

A total of 1334 sandflies were collected from sheep pens in two villages in Jiexiu City, Shanxi Province, on 29 July 2022, with 1283 from Baishui village and 51 from Guobi village. The collected sandflies were grouped into 45 pools for the detection of the COI gene. Subsequent analysis combined with the results of morphological identification revealed that all of the sandflies belonged to the species *P*. *chinensis*.

### 3.2. Isolation and Identification of Isolated WUXV Strains

A total of 45 pools of ground sample supernatants were inoculated into BHK-21 and C6/36 cells, respectively. After three consecutive generations of cultivation, only three samples—SXJX2239, SXJX2241, and SXJX2243—induced transmissible CPEs in BHK-21 cells. Compared with normal BHK-21 cells, infected BHK-21 cells exhibited slower growth, rounding, and detachment (Figure 2). The qRT-PCR assay for the specific detection of WUXV yielded positive results for all three viral strains. None of the samples showed CPEs after three generations of culture in C6/36 cells.

### 3.3. Viral Sequencing Results

The entire genomes of three newly identified WUXV strains (SXJX2239, SXJX2241, and SXJX2243), along with previously maintained strains (11 from Wuxiang County and 17 from Yangquan City), were sequenced using the Nova-Seq platform. This resulted in an average of 12,784,437.2 reads per sample (range: 7,779,288 to 24,134,504). Approximately 87.55% of reads in each sample passed the quality check and then were successfully mapped to the reference sequences. The consensus sequence for each segment in each sample was generated at a coverage depth of 20× (Appendix A).

### 3.4. Phylogenetic Analysis and Genetic Distance

Viral phylogenetic analysis of the nucleotide sequences of 33 WUXV strains showed that regardless of the gene segment, WUXV strains previously isolated in Wuxiang County and Yangquan City clustered together and formed clade 1, whereas the novel strains isolated in Jiexiu City in the current study formed a separate evolutionary branch, termed clade 2 (Figure 3). Notably, the topology of the phylogenetic analysis aligned with the virus isolation locations, specifically in the L segment of clade 1 (Figure 3A). In the M segment of clade 1, strains SXYQ1968-2, SXYQ1968-3, and SXYQ1941-5 clustered more closely with strains from Wuxiang County than with those from Yangquan City (Figure 3B). The same was observed for strains SXYQ1968-2, SXYQ1968-3, SXYQ1922-2, SXYQ1966-4, and SXYY1966-5 in the S segment of clade 1 (Figure 3C). The nucleotide (amino acid) homology of the L, M, NS, and N segments between the two clades was 89.9–90.7% (98.5–98.8%), 84.9–87.8% (89.2–92.4%), 87.6–90.0% (93.9–95.8%), and 90.6–92.3% (98.8–99.6%), respectively (Appendix A).

### 3.5. Reassortment/Recombination Analysis

No reassortment or recombination events occurred between the two clades. However, four recombination events were detected in clade 1 involving strains SXWX1920-3, SXWX1913-2, SXYQ1941-5, and SXYQ1966-4 (Table 1). The SXYQ1941-5 parents were obtained from Yangquan City and Wuxiang County, China. The above results were further verified using SimPlot v3.5.1 (Appendix A).

### 3.6. The Amino Acid Mutation and the Influence of Protein Tertiary Structure

A comparison of amino acid sequences between the two clades of WUXV identified eighty-seven distinct amino acid mutations, with seventeen, sixty, nine, and one substitution(s) in the proteins RdRp, M, NS, and N, respectively (Appendix A). Subsequent selection pressure analysis revealed that 17 sites of the WUXV genome were under positive selection pressure, with two, thirteen, one, and one site(s) in the L, M, NS, and N segments, respectively (Table 2). Notably, among the mutation sites between the two clades, M-312 and M-340 also exhibited positive selection pressure. The tertiary structure of protein M of strains SXWX1813-2 (on behalf of clade 1) and SXJX2241 (on behalf of clade 2) was predicted, and the alignment of the two proteins resulted in a high degree of similarity with an RMSD value of 0.784. Furthermore, the two proteins appeared to have identical tertiary loop structures at sites M-312 and M-340 (Figure 4).

## 4. Discussion

WUXV, a novel *Phlebovirus* isolated from sandflies (*P. chinensis*) in China, was confirmed as the local viral population in Wuxiang County and Yangquan City, Shanxi Province. In this study, WUXV was detected in sandflies from Jiexiu City. Phylogenetic analysis of the WUXV genome sequences revealed two distinct viral clades with geographical differences.

Several important insect-borne pathogens are endemic to Shanxi Province [34,35,36,37], including the Japanese encephalitis virus, Banner virus, and *Leishmania donovani*. Among these, mosquito-borne Japanese encephalitis and sandfly-borne kala-azar have caused public health incidents. This has raised concerns regarding the threat of insect-borne pathogens to the health of people in Shanxi Province. In 2018, two novel viruses were isolated from sandflies collected from Shanxi Province [10,38]. Both belong to the *Phlebovirus* genus and were designated as WUXV and Hedi virus (HEDV). Notably, both WUXV and HEDV replicated in the BHK-21 cells; however, HEDV did not cause CPE. The cellular growth characteristics of the three novel WUXV strains in this study mirrored those of previously isolated WUXV strains, suggesting potential mammalian adaptation capabilities. Thus, the evolutionary characteristics of WUXV should be noted.

The evolution of WUXV exhibits geographical features. In this study, the distribution of WUXV expanded from Wuxiang County and Yangquan City to Jiexiu City. According to the phylogenetic trees generated in this study, WUXV could be divided into two clades. Clade 1 contained the strains isolated from Wuxiang County and Yangquan City, whereas clade 2 contained the strains isolated from Jiexiu City. Geographically related evolutionary traits are not unique to WUXV; similar patterns exist for other arboviruses. For instance, severe fever with thrombocytopenia syndrome virus (SFTSV) and Crimean–Congo hemorrhagic fever virus (CCHFV), both also members of the *Bunyavirales* order, demonstrate such traits. SFTSV can be categorized into Chinese and Japanese clades, whereas CCHFV can be categorized into three African clades, one Asian clade, and two European clades based on the geographical distribution of the virus strains [39,40]. The findings on WUXV clades suggest that the environment may have influenced the evolution of WUXV [41]. However, more WUXV strains from different locations are needed to further investigate whether other viral clades exist and determine the origins of WUXV. Therefore, nationwide surveillance of WUXV is necessary.

No instances of gene reassortment/recombination were observed between the two clades; however, they occurred within clade 1. No gene exchange occurred between the two clades, indicating that WUXV evolved independently in different locations. Gene reassortment/recombination within clade 1 also suggested that the strains from Yangquan City and Wuxiang County were more closely related to each other. The gene reassortment/recombination of most bunyaviruses involves M-segment replacement [42,43]. The current study found that gene exchange in WUXV occurred mainly in the M segment. One of the reassortants identified in this study, strain SXYQ1941-5, was generated by combining the S and L segments from strain SXYQ1927-2 with the M segment from strain SXWX1923-4. This reassortment event may explain why strain SXYQ1941-5 clustered more closely with the strains from Wuxiang County in the M segment.

Total of eighty-seven clade-specific mutations were present between the clades, with seventeen, sixty, nine, and one site(s) in the RdRp, M, NSs, and N proteins, respectively. RdRp and N can form a ribonucleoprotein complex (RNP), which is responsible for the replication of WUXV [16]. The M protein plays a crucial role in the entry into target cells and the assembly of progeny particles. The function of the NSs protein in WUXV remains elusive. Nonetheless, the NSs protein is widely recognized as a pivotal virulence determinant among phleboviruses, exerting significant influence on the evasion strategies employed by the virus to counteract the host’s innate immune response [22,44]. Among the mutation sites mentioned above, only two (M-312 and M-340) in the M segment exhibited both amino acid mutations and positive selection pressure. Mutations at sites M-312 and M-340 did not alter the protein structure significantly, based on protein structural predictions. However, even a single amino acid can significantly affect the function of an encoded protein. For example, the N342K mutation in neuraminidase likely enhances influenza B virus replication and pathogenicity [45]. RVFV, a virus of the same genus as WUXV, exhibits significantly increased virulence through the S39C/S40C mutation in the NSs protein [46]. Further research is needed to determine whether sites M-312 and M-340 affect the function of protein M or whether they have other biological significances.

This study involved a genetic analysis of WUXV to elucidate its genetic characteristics. Currently, surveillance of WUXV in China is in its early stages; thus, the WUXV strains examined in this study were limited to Shanxi Province. Moreover, the virus isolation dates between strains were intermittent. Therefore, nationwide surveillance of WUXV should be conducted continuously in the future to gain a deeper and more comprehensive understanding of the origin of WUXV and its overall distribution in China.

## 5. Conclusions

In this study, three novel WUXV strains were isolated in Jiexiu City. Subsequently, we found that the virus belonged to two clades. No reassortment or recombination events were detected between clades, and 87 amino acid mutation sites were identified. In addition, there were 17 positively selected sites in WUXV, and sites M-312 and M-340 were both mutated and positively selected. Additionally, this study provides an update on the distribution of WUXV in China, highlighting the need for continuous nationwide surveillance of WUXV in the future.

## Figures and Tables

**Figure 1 viruses-16-00103-f001:**
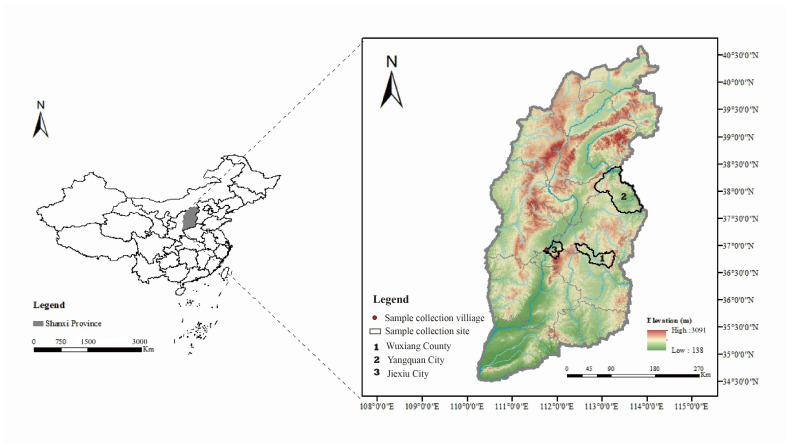
Sample collection villages in Jiexiu City, Shanxi Province, China, 2022. The area shaded by gray in the figure is Shanxi Province in central China. The areas outlined in black in the figure are sample collection sites of WUXV. Among them, sample collection sites 1 (Wuxiang County) and 2 (Yangquan City) are previous WUXV isolation locations. Sample collection site 3 (Jiexiu City) is the sample collection site of this study, and the red circles indicate the sample collection villages.

**Figure 2 viruses-16-00103-f002:**
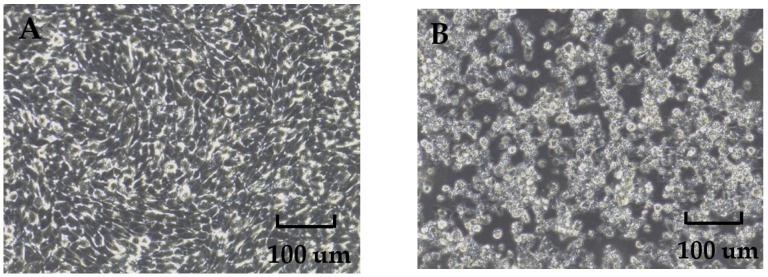
Cytopathic effect of the strain SXJX2241 in BHK-21 cells. (**A**) Normal BHK-21 cells (5 days). (**B**) Strain SXJX2241 caused a cytopathic effect in BHK-21 cells (5 days). Magnification: 200×.

**Figure 3 viruses-16-00103-f003:**
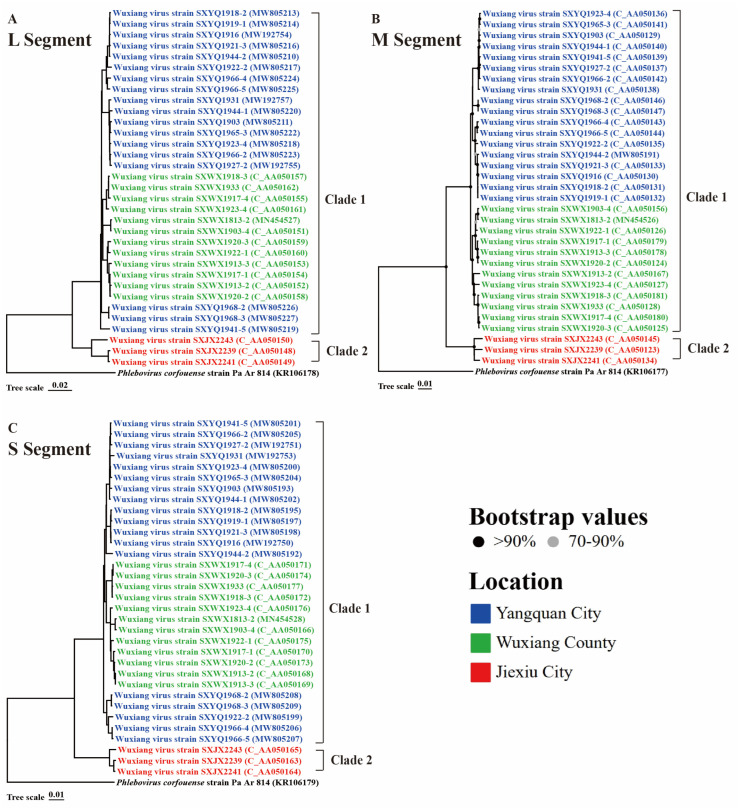
Molecular phylogenetic analysis of WUXV genome. (**A**) Phylogenetic analysis of L segment sequence of WUXV. (**B**) Phylogenetic analysis of M segment sequence of WUXV. (**C**) Phylogenetic analysis of S segment sequence of WUXV. MEGA v7.0.26 was used to perform the phylogenetic analysis, using the neighbor-joining method with a bootstrap value of 1000, and *Phlebovirus corfouense* was regarded as an outgroup.

**Figure 4 viruses-16-00103-f004:**
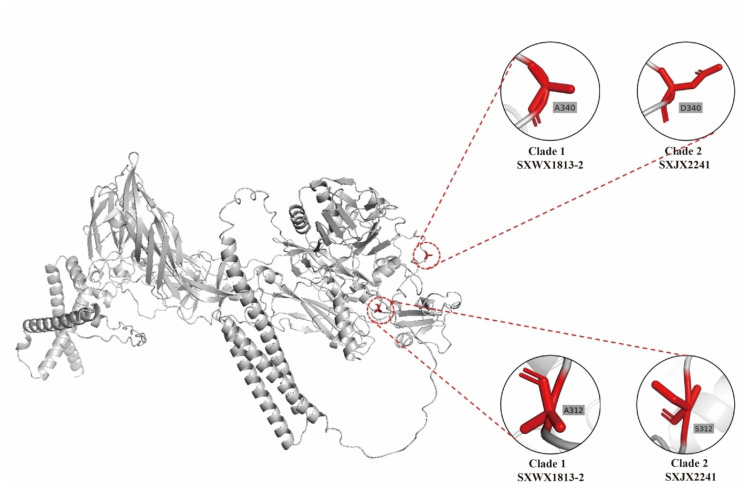
The predicted tertiary structures of the M protein of WUXV. The zoomed-in areas indicate the tertiary structures at sites 312 and 340 in clade 1 (strain SXWX1813-2) or clade 2 (strain SXJX2241).

**Table 1 viruses-16-00103-t001:** Recombination events detected in clade 1 strains.

Recombinant Sequence	Minor Parental Sequence ^1^	Major Parental Sequence ^2^	Begin of Recombination (nt)	End of Recombination (nt)	Detection Methods						
					RDP	GENECONV	Bootscan	Maxchi	Chim- aera	SiScan	3Seq
SXWX1920-3	SXWX1922-1	SXWX1917-4	6261	10,365	2.96 × 10^−14^	2.96 × 10^−12^	3.72 × 10^−10^	1.03 × 10^−19^	2.51 × 10^−18^	2.28 × 10^−15^	1.68 × 10^−34^
SXWX1913-2	SXWX1920-2	SXWX1813-2	6258	11,973	1.15 × 10^−14^	2.14 × 10^−15^	4.44 × 10^−9^	4.72 × 10^−15^	7.83 × 10^−11^	4.72 × 10^−20^	1.22 × 10^−22^
SXYQ1941-5	SXWX1923-4	SXYQ1927-2	6247	10,380	3.24 × 10^−19^	1.20 × 10^−14^	1.83 × 10^−12^	2.50 × 10^−17^	5.10 × 10^−9^	5.57 × 10^−32^	8.58 × 10^−5^
SXYQ1966-4	SXYQ1965-3	SXYQ1966-5	8943	9892	2.19 × 10^−16^	3.12 × 10^−11^	2.18 × 10^−16^	2.70 × 10^−7^	1.78 × 10^−5^	8.50 × 10^−4^	3.30 × 10^−12^

Note: ^1^ Parent contributing the smaller fraction of sequence; ^2^ Parent contributing the larger fraction of sequence.

**Table 2 viruses-16-00103-t002:** Amino acid sites under positive selection pressure in the WUXV genome.

Segment	Codon	Method							
		FEL		FUBAR		MEME		SLAC	
		dN-dS	*p*-Value	dN-dS	Pos.pro *	ω^+^	*p*-value	dN-dS	*p*-Value
L	237	/	/	/	/	112.475	0.076	/	/
	744	7.760	0.037	15.411	0.977	43.582	0.012	/	/
M	126	/	/	/	/	91.747	0.097	/	/
	149	/	/	3.610	0.907	/	/	/	/
	182	/	/	/	/	132.870	0.081	/	/
	212	/	/	/	/	20.188	0.056	/	/
	298	/	/	/	/	45.696	0.040	/	/
	312	2.003	0.090	/	/	/	/	/	/
	340	/	/	/	/	426.456	0.098	/	/
	437	/	/	/	/	76.178	0.098	/	/
	566	/	/	/	/	13.571	0.078	/	/
	817	/	/	/	/	318.453	0.065	/	/
	1234	/	/	/	/	536.459	0.035	/	/
	1289	3.343	0.047	/	/	3.315	0.065	/	/
	1358	/	/	/	/	125.278	0.071	/	/
NS	30	/	/	10.461	0.962	/	/	/	/
N	107	/	/	27.425	0.998	157.352	0.071	/	/

Note: * Posterior probability, ‘/’ indicates no significant signal is detected in positive selection; ‘dN’ indicates rate of non-synonymous substitutions; ‘dS’ indicates rate of synonymous substitutions; **‘ω^+^’** indicates the value of dN^+^/dS.

## Data Availability

The data presented in this study are openly available in GenBase (https://ngdc.cncb.ac.cn/genbase/, accessed on 13 November 2023).

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
