# Peer review of "Genetic Characteristics of Wuxiang Virus in Shanxi Province, China"

_viruses, 2024, doi:10.3390/v16010103_

Round 1

Reviewer 1 Report

Comments and Suggestions for Authors

The manuscript describes the isolation of three strains of WUXV phlebovirus from sandflies, the whole genome sequence of 14 strains (including the 3 new isolated strains) and 17 strains L-segment sequences. A phylogenetic analysis is performed enabling the identification of two clusters that can be related to the strains geographic origin.  Nucleotide and aa sequences are also compared as well as, a reassortment/recombination  analysis.

The amount of genomic sequence data is soundness and deserves to be published, however the data presentation and the manuscript text could be improved.  

Specifically, I have the following comments:

Authors: I believe an extra “and” should be removed after last author name Huanyu Wang

Abstract (lines 26-27): Suggestion the sentence “However, four reassortments or recombination events occurred in clade 1.” should be changed to “However, four reassortments or recombination events could be detected in clade 1 strains.

Materials and Methods (lines 92-93): C3/36 cells are referred as Aedes albopictus egg cells. To my knowledge C6/36 cells were isolated from Aedes albopictus larva tissues, not eggs. Is the cell line used different? – If not, the sentence should be changed to “Aedes albopictus cells (C6/36 cells)…”. Please note that Aedes albopictus should be in italic.

Figure 1 : The yellow outline and the yellow circles are very difficult to see. Please choose another colour/way to highlight the locations

Sandflies collection (line 150):  the identification method/ taxonomic characteristic used to identify sandflies as P. chinensis should be described. Identification by experienced entomologists is not a scientific description of the identification procedure. Moreover, this description should be place in Materials and Methods section.

Figure 3: The presented phylogenetic trees are too small to be seen. Different colour sequences are not clarified in the figure legend.

Figure 3 legend (lines 189-190): Phlebovirus corfouense should be in italic and P in capital letter

Table 1 title: suggestion, change to “Recombination events detected in clade 1 strains. “

Figure 4 legend: the tertiary structures are at sites 312 and 340 not 314 and 340?

Discussion (lines 224-225): The sentence “Gene expression analysis of WUXV revealed two distinct viral clades, with geographical differences” does not describes the presented work. No gene expression analysis were performed in this study. The sentence should be rephrased.

Suggestion “Phylogenetic analysis of WUXV genome sequences revealed two distinct viral clades, with geographical differences.”

Table S1: please correct refference to reference

Table S4- Not NS segment but Ns gene

Table S5 – Not N segment but N gene

Table S6 – the first column is not only referring to proteins, but proteins and genome segments (M), please correct

Comments on the Quality of English Language

Table S1: please correct refference to reference

Reviewer 2 Report

Comments and Suggestions for Authors

The manuscript provides an interesting analysis of new WUXV isolates that belong to a different clade. The text is well written and documented.

Minor comments:

Recombination events: the authors could indicate the approximate sites in the different examples. A comment should be given why the PhylPro method did not result in detection.

Table 2: explain „dN-dS“, „ω”. It needs to be commented why the different methods indicate different sites. Notably, sites M-312 and M-340 are identified by different methods. There is limited overlap between the different methods while no sites were identified by SLAC. Please explain the value of the analysis using different methods.

Line 224: should read: “Phylogenetic analysis of WUXV revealed two distinct viral clades”.
